# Protective Effects of 3′-Epilutein and 3′-Oxolutein against Glutamate-Induced Neuronal Damage

**DOI:** 10.3390/ijms241512008

**Published:** 2023-07-26

**Authors:** Ramóna Pap, Edina Pandur, Gergely Jánosa, Katalin Sipos, Ferenc Rómeó Fritz, Tamás Nagy, Attila Agócs, József Deli

**Affiliations:** 1Department of Pharmaceutical Biology, Faculty of Pharmacy, University of Pécs, Rókus u. 2, H-7624 Pécs, Hungary; pap.ramona@pte.hu (R.P.); edina.pandur@aok.pte.hu (E.P.); janosa.gergely@gytk.pte.hu (G.J.); katalin.sipos@aok.pte.hu (K.S.); ferkofritz@gmail.com (F.R.F.); 2Department of Laboratory Medicine, Faculty of Medical Sciences, University of Pécs, Ifjúság út 13, H-7624 Pécs, Hungary; nagy.tamas@pte.hu; 3Department of Biochemistry and Medical Chemistry, Medical School, University of Pécs, Szigeti út 12, H-7624 Pécs, Hungary; attila.agocs@aok.pte.hu; 4Department of Pharmacognosy, Faculty of Pharmacy, University of Pécs, Rókus u. 2, H-7624 Pécs, Hungary

**Keywords:** 3′-epilutein, 3′-oxolutein, oxidative stress, glutamate, neuron

## Abstract

Dietary lutein can be naturally metabolized to 3′-epilutein and 3′-oxolutein in the human body. The epimerization of lutein can happen in acidic pH, and through cooking, 3′-epilutein can be the product of the direct oxidation of lutein in the retina, which is also present in human serum. The 3′-oxolutein is the main oxidation product of lutein. Thus, the allylic oxidation of dietary lutein can result in the formation of 3′-oxolutein, which may undergo reduction either to revert to dietary lutein or epimerize to form 3′-epilutein. We focused on the effects of 3′-epilutein and 3′-oxolutein itself and on glutamate-induced neurotoxicity on SH-SY5Y human neuroblastoma cells to identify the possible alterations in oxidative stress, inflammation, antioxidant capacity, and iron metabolism that affect neurological function. ROS measurements were performed in the differently treated cells. The inflammatory state of cells was followed by TNFα, IL-6, and IL-8 cytokine ELISA measurements. The antioxidant status of the cells was determined by the total antioxidant capacity kit assay. The alterations of genes related to ferroptosis and lipid peroxidation were followed by gene expression measurements; then, thiol measurements were performed. Lutein metabolites 3′-epilutein and 3′-oxolutein differently modulated the effect of glutamate on ROS, inflammation, ferroptosis-related iron metabolism, and lipid peroxidation in SH-SY5Y cells. Our results revealed the antioxidant and anti-inflammatory features of 3′-epilutein and 3′-oxolutein as possible protective agents against glutamate-induced oxidative stress in SH-SY5Y cells, with greater efficacy in the case of 3′-epilutein.

## 1. Introduction

The process of oxidative stress is characterized by disturbances in the balance between the production of reactive oxygen species (ROS) and the capacity of detoxification in cells. In addition, oxidative stress has become a pivotal factor in several physiological and pathological processes. Neurons are especially susceptible to oxidative damage according to their large dependence on oxidative phosphorylation for energy production. Recently, extensive research has been focused on understanding the effects of oxidative stress on neuronal function and, thus, its consequences in neurodegenerative diseases [1,2]. 

Carotenoids are bioactive compounds that are used in multipurpose biological roles according to their potential antitumoral, immunomodulatory, anti-inflammatory, antibacterial, and neuroprotective effects, among others. Thus, carotenoids have been at the forefront of research for their possible protective effects against neuronal damage induced by oxidative stress. In addition, their anti-inflammatory features seem to be able to mitigate the inflammatory response and contribute to reducing neuronal damage [3,4,5]. Lutein ((3R,3′R,6’R)-β,ε-carotene-3,3′-diol) is a xanthophyll carotenoid well known for its benefits in eye health and anti-inflammatory features [6]. Lutein acts as an antioxidant by donating an electron to neutralize free radicals. This process oxidizes lutein itself, forming an oxidized molecule. However, other antioxidants, such as vitamins C and E, help regenerate lutein back to its active form, allowing it to continue protecting cells from oxidative damage [7,8].

Dietary lutein can metabolize into 3′-epilutein and 3′-oxolutein. Both 3′-epilutein and 3′-oxolutein are structurally related to lutein and belong to the xanthophyll class of carotenoids. The 3′-epilutein ((3R,3′S,6′R)-β,ε-carotene-3,3′-diol) can be metabolized via epimerization of lutein in acidic pH and through cooking as well as 3′-epilutein can be the product of the direct oxidation of lutein in retina besides can be extracted from the petals of Caltha palustris. The addition of lutein in combination with 3′-epilutein has been shown to increase the macular pigment optical density in early-stage AMD [9,10,11]. The 3′-oxolutein (3-hydroxy-β,ε-carotene-3′-one) is the main oxidation product of lutein, which can be formed due to a hydroxyl group at the C-3′ group which could have been activated by the neighboring allylic double bond. Its presence has been reported with increasing levels in human development, in arterial cord blood of infants in ocular tissues as well [12,13]. The bioavailability of the derivates of lutein might be different; thus, their function can also alter. The suspected antioxidant or anti-inflammatory effects of these compounds have not been investigated in neuronal cells. 

Antioxidant molecules are able to reduce oxidative stress-induced damage by neutralizing the produced ROS. The phenomena of oxidative stress and total antioxidant capacity (TAC) play a major role in neurons and the development of neurodegenerative diseases. TAC provides a characterization of the disequilibrium between pro- and antioxidants and is responsible for the protection against oxidative stress. The reduction in TAC or antioxidant protection may contribute to the development of neurodegenerative diseases [14,15,16]. 

Neurons have a comprehensive antioxidant defense system to combat the damaging effects of oxidative stress. Antioxidant enzymes play a central role in the mechanisms of scavenging. The antioxidant enzyme system includes Superoxide dismutase (SOD), which composes the front line of the antioxidant enzyme system against oxidative stress-mediated damage [17,18,19]. Heme oxygenase-1 (HO-1) is an important antioxidant enzyme and thus plays a central role in the degradation of heme and the subsequent production of iron. HO-1 also exhibits anti-inflammatory and protective properties in neurons besides being reported to mediate protective or detrimental effects through the induction of ferroptosis [20,21].

Ferroptosis contributes to the regulation of oxidative stress and the inflammatory responses in disease. The process is regulated by iron and characterized by lipid peroxidation, which is caused by a disruption in the metabolism of lipids, antioxidants such as glutathione, and iron and can accelerate acute central nervous system injury [22,23,24]. The uptake of iron is regulated by Transferrin receptor-1 (TfR1), while Ferritin heavy chain (FTH) is related to the storing of the recruited iron. Ferroportin (FPN) is the only known transmembrane exporter of nonheme iron. The accumulation of a high amount of iron and the dysregulation of the iron metabolism is associated with the development of neurodegenerative diseases also characteristics of ferroptosis [25,26,27,28]. 

The organic compound thiol plays a major role in maintaining the redox balance through the direct reaction with reactive molecules via their sulfhydryl group [29]. In lipid peroxidation, reactive lipids can interact with thiol compounds, resulting in the formation of oxidized thiol species and the generation of disulfide bonds [30,31]. 

Oxidative stress affects neurons through the generation of proinflammatory cytokines such as Tumor necrosis factor alpha (TNFα), Interleukin (IL)-6 (IL-6), or IL-8 as a response to the cellular damage induced by ROS. These cytokines spread the inflammatory response as well as incorporate neuronal dysfunction or cell death [32,33]. 

As an excitatory neurotransmitter in CNS, glutamate plays a crucial role in memory, synaptic plasticity, learning, motor function, and neural transmission [34]. Excess glutamate is a critical pathogenic event that induces brain disorders such as Alzheimer’s disease and Parkinson’s disease [35,36].

This study focuses on the potential protective effects of the xanthophyll carotenoids 3′-epilutein and 3′-oxolutein on oxidative stress in glutamate-induced neurotoxicity in SH-SY5Y cells. Although these natural compounds can be found in the human body, their effects on neuronal cells have not been investigated yet. Our results revealed first the anti-inflammatory and antioxidant effects of both 3′-epilutein and 3′-oxolutein in SH-SY5Y cells. 

## 2. Results

### 2.1. The 3′-Epilutein and 3′-Oxolutein Suppresses the Glutamate-Induced ROS in SH-SY5Y Cells

Excessive production or insufficient removal of ROS can lead to oxidative stress. According to the results of the viability assays, 10 ng/µL of 3′-epilutein and 3′-oxolutein was used in the treatments. The effects of 3′-epilutein and 3′-oxolutein in SH-SY5Y cells were first followed through the ROS production and determined upon a glutamate-induced oxidative stress condition. The oxidative stress was induced with 1 mM or 5 mM of glutamate in the cells, and concentrations were selected by our previous research. Treatment with both 3′-epilutein and 3′-oxolutein significantly decreased the ROS elevated by glutamate (Figure 1). 

### 2.2. The 3′-Epilutein Increases the Total Antioxidant Capacity in SH-SY5Y Cells

Antioxidants play a pivotal role in maintaining cellular health by scavenging harmful radicals and eliminating the generated ROS. Next, colorimetric total antioxidant capacity (TAC) measurements were performed to follow the antioxidant status of the cells. Treatment with 5 mM glutamate significantly decreased the small molecule antioxidants level at 24 h, 48 h, and 72 h. SH-SY5Y cells treated with 3′-epilutein showed elevation in the small molecule antioxidants level at 24 h, 48 h, and 72 h. Treatment with 3′-epilutein could restore the small molecule antioxidant level in the glutamate-treated cells. The 3′-oxolutein exerted increasing effects on the small molecule antioxidants at 48 h but could not modify the TAC levels in glutamate treatments. Besides small molecule antioxidants, a rise was detected in the protein antioxidants level in the case of the EG5 treatment at 48 h and 72 h (Figure 2). 

### 2.3. The Attenuating Effects of 3′-Epilutein and 3′-Oxolutein on SOD Activity in Glutamate-Induced SH-SY5Y Cells

Besides its anti-inflammatory features, SOD composes the front line in the defense against oxidative stress-mediated damage; thereafter, SOD enzyme activity measurements were performed. Glutamate treatment significantly decreased the SOD activity in the cells. Glutamate treatment decreased SOD enzyme activity, whereas each 3′-epilutein-treated SH-SY5Y cell showed increased SOD levels at all time points. The 3′-epilutein treatment compensated for the effect of glutamate in both EG1 and EG5 treatments at 24 and 48 h. The 3′-oxolutein treatment alone did not alter SOD activity; however, it did not inhibit the decrease in SOD in glutamate-treated cells (Figure 3).

### 2.4. The 3′-Epilutein and 3′-Oxolutein Modulates the Expression of Ferroptosis-Related Iron Regulatory Genes and the Iron Content in Glutamate-Induced SH-SY5Y Cells

Oxidative stress and ferroptosis are intricately linked processes that can have profound implications for neuronal health since oxidative stress can lead to the accumulation of ROS, which can trigger the ferroptotic cascade in neurons. Therefore, the expression of Ferritin heavy chain (FTH), Transferrin receptor 1 (TfR1), Ferroportin (FPN), and Heme oxygenase-1 (HO-1) genes were examined due to their roles in iron metabolism and ferroptosis. 

The relative mRNA expression of TfR1 and FTH was significantly increased by 5 mM glutamate treatment in all three treatment periods, while FPN expression showed a slight but not significant decrease suggesting the uptake and retention of iron in the glutamate-treated SH-SY5Y cells (Figure 4). 

Neither 3′-epilutein nor 3′-oxolutein treatments alone did not alter the relative mRNA expression of TfR1, FTH, and FPN. In EG1 and EG5 treatments, 3′-epilutein reduced mRNA expression in TfR1 and FTH coding genes compared to the effects of glutamate. On the other hand, the OG1 and OG5 treatments did not cause significant alterations in TfR1 and FTH relative mRNA expressions compared to the glutamate treatments (Figure 4).

The relative mRNA expression of HO-1 was elevated upon glutamate treatments at 24 h and 48 h with a decreasing tendency and returned to the control level at 72 h. Treatment with 3′-epilutein and 3′-oxolutein did not modulate the relative mRNA expression of HO-1. The expression of HO-1 was reduced in EG1 and EG5 treatments compared to glutamate treatments, while administration of OG1 and OG5 showed similar expression levels as glutamate-treated SH-SY5Y cells (Figure 4). 

These results suggest that 3′-epilutein was able to modulate the expression of genes regulating iron metabolism in glutamate-induced stress while 3′-oxolutein did not.

The cellular level of iron plays a crucial part in the occurrence of ferroptosis and iron-dependent lipid peroxidation. Thus, total iron measurements were performed in the SH-SY5Y cells. 

In the glutamate-treated cells, the content of iron was increased at 24, 48, and 72 h. Treatment with 3′-epilutein and 3′-oxolutein alone did not modulate the level of iron in SH-SY5Y cells. A decrease was measured in the iron content in EG5 and OG5 treatments for all three treatment durations, as well as in EG1 and OG1 treatments at 72 h (Figure 5). 

### 2.5. Downregulatory Effects of 3′-Epilutein and 3′-Oxolutein in the Glutamate-Induced Increase of Gene Expressions toward Lipid Peroxidation

Ferroptosis is a specific form of programmed cell death induced by iron-dependent lipid peroxidation. In the context of ferroptosis and lipid peroxidation, the expression levels of specific genes, such as ALOX5, ALOX15, and ACSL4, are of particular interest. ALOX5 and ALOX15 are genes involved in the metabolism of polyunsaturated fatty acids and synthesis of lipid mediators and inflammatory processes, while ACSL is a critical player in ferroptosis and lipid peroxidation according to its role in facilitating the incorporation of specific polyunsaturated fatty acids into phospholipids within cell membranes. 

The 5 mM glutamate treatment increased the relative mRNA expression of ALOX5, ALOX15, and ACSL4 coding genes in 24, 48, and 72 h treatments. Treatment with 3′-epilutein and 3′-oxolutein did not alter the expression of ALOX5, ALOX15, and ACSL4 genes compared to controls. Both 3′-epilutein and 3′-oxolutein reduced the relative mRNA expressions of ALOX5, ALOX15, and ACSL4 genes in EG1, EG5, OG1, and OG5 treatments compared to glutamate-treated cells with a greater impact in the case of 3′-epilutein (Figure 6). 

These results may suggest that both carotenoids could play a role in the prevention of lipid peroxidation induced by glutamate.

### 2.6. Elevation of Thiol Compounds upon 3′-Epilutein and 3′-Oxolutein in SH-SY5Y Cells

The regulatory interplay between lipid peroxidation and the quantity of thiol compounds is tightly regulated through the balance of oxidative and antioxidative processes within cells. The antioxidant thiol compounds play a pivotal role in the prevention of oxidative stress-induced impairment and also in the protection of the cells against oxidative stress. Therefore, thiol content determinations were performed in the SH-SY5Y cells. 

Thiol levels were significantly reduced in SH-SY5Y cells treated with 5 mM glutamate after 24, 48, and 72 h, in addition to the cells treated with 1 mM glutamate after 48 and 72 h. Treatment with 3′-epilutein elevated thiol content in all three treatments, while 3′-oxolutein significantly increased thiol levels after 48 and 72 h. Both 3′-epilutein and 3′-oxolutein could increase the thiol levels in EG1, EG5, OG1, and OG5 treatments compared to glutamate treatments. In addition, 3′-epilutein increased further thiol levels at 48 and 72 h compared to the control (Figure 7). 

The decreased thiol levels upon glutamate treatments are possible indicators of the impairment in the antioxidant defenses; thus, 3′-epilutein and 3′-oxolutein seem to be able to mitigate the glutamate’s effect on thiol compounds.

### 2.7. Effects of 3′-Epilutein and 3′-Oxolutein on Proinflammatory Cytokine Secretion Induced by Glutamate in SH-SY5Y Cells 

Oxidative stress and inflammation are crucial participants in the neuronal regulation of health and diseases. Oxidative stress can stimulate the inflammatory response in cells and increase the production of inflammatory cytokines, while inflammatory cytokines stimulate ROS production, which can lead to further oxidative damage. Therefore, IL-6, IL-8, and TNFα ELISA were investigated to examine the secretions of the proinflammatory cytokines. 

#### 2.7.1. Secretion of IL-6

First, IL-6 measurements were performed in SH-SY5Y cells. The secretion of IL-6 was significantly increased in glutamate-treated cells at 24, 48, and 72 h compared to controls. Treatment with 3′-epilutein reduced the IL-6 secretion in the cells after 48 and 72 h, while 3′-oxolutein did not induce alteration in SH-SY5Y compared to controls. In EG1 and EG5 treatments, 3′-epilutein significantly decreased the IL-6 levels (Figure 8). 

#### 2.7.2. Secretion of IL-8

The dysregulation of IL-8 cytokine links to systemic inflammation. Next, IL-8 cytokine ELISA was investigated. 

IL-8 secretion was elevated in glutamate-treated cells. Treatment with 3′-epilutein decreased the IL-8 levels at 48 h, while 3′-oxolutein alleviated the IL-8 levels at 24 and 48 h compared to controls. Both 3′-epilutein and 3′-oxolutein treatments reduced the glutamate-induced IL-8 secretions in the EG1, EG5, OG1, and OG5 treatments (Figure 9).

#### 2.7.3. Secretion of TNFα

The increased presence of the proinflammatory cytokine TNFα in cells may contribute to oxidative stress by increasing ROS production and promoting oxidative damage. Glutamate increased the TNFα levels in 24, 48, and 72 h compared to controls. Treatment with 3′-epilutein decreased the TNFα secretion at 24 h and did not modulate the TNFα levels at 48 and 72 h. Meanwhile, treatment with 3′-oxolutein did not alter the TNFα secretions in SH-SY5Y cells. Both 3′-epilutein and 3′-oxolutein diminished the TNFα levels in EG1, EG5, OG1, and OG5 treatments compared to the glutamate-treated cells (Figure 10). 

These results suggest that 3′-epilutein and 3′-oxolutein alter the secretion of inflammatory cytokines in glutamate-induced oxidative stress; both seem to be able to attenuate the levels of IL-6, IL-8, and TNFα proinflammatory cytokines.

## 3. Discussion

The importance of the xanthophyll carotenoid lutein in visual health, mainly in the improvement of AMD and in light-induced oxidative stress, is well known. In addition, lutein has been shown to benefit cognitive function and heart health. Lutein has been reported to have antioxidant and anti-inflammatory features [6]. Dietary lutein can be metabolized to 3′-epilutein and 3′-oxolutein, among others. The 3′-epilutein is a natural metabolite of lutein in the human body. The epimerization of lutein occurs in acidic pH and through cooking, as well as 3′-epilutein can be the product of the direct oxidation of lutein in the retina. The presence of the 3′-epimer of lutein has been reported in human serum and ocular tissues. The addition of lutein in combination with 3′-epilutein has been shown to increase the macular pigment optical density in early-stage AMD [10,37,38,39]. The 3′-oxolutein is the main oxidation product of lutein, which can be formed due to a hydroxyl group at the C-3′ group which could have been activated by the neighboring allylic double bond. Its presence has been reported in human serum, preterm and term infants’ arterial cord blood, and ocular tissues [40]. The allylic oxidation of dietary lutein can result in the formation of 3′-oxolutein, which may undergo reduction either to revert to dietary lutein or else epimerize at C-3′ to form 3′-epilutein [11]. The bioavailability of the derivates of lutein might be different; thus, their potential effects can also alter. 

Oxidative stress is a condition in which the production of ROS exceeds the capacity of the cell’s antioxidant defense system. ROS present in neurons plays an important role in the normal functioning of cells and the development of various pathological conditions of the nervous system. The process can contribute to neuronal damage and the development of neurodegenerative diseases such as Alzheimer’s, Parkinson’s, and ischemic brain injury [41,42,43]. Glutamate is the most abundant excitatory neurotransmitter in the brain that plays a major role in neuronal communication. It participates in processes such as learning, memory, and cognition [44]. Oxidative stress induced by glutamate is a process where high glutamate levels in cells have a detrimental effect on antioxidant defenses. The mechanism of glutamate-induced oxidative stress is a diverse and concentration-dependent procedure, also the major mechanism of glutamate-induced neurotoxicity [45,46,47]. We examined the effects of 3′-epilutein and 3′-oxolutein on ROS production in glutamate-induced oxidative stress in SH-SY5Y cells. Both carotenoids significantly reduced the intracellular ROS elevated by glutamate, suggesting their modifying effect on oxidative stress. 

The antioxidant capacity of neurons is important according to their exposition to high levels of oxidative stress. Maintaining an optimal TAC is important for protecting neurons and promoting overall health; higher TAC levels are associated with better cognitive function, reduced risk of neurodegenerative diseases, and overall improved health outcomes [31,48,49,50]. In our results, the total antioxidant capacity showed a reduction in small molecule antioxidants upon 5 mM glutamate treatment, while 3′-epilutein elevated it in each treatment alone as well as in the presence of glutamate as a mitigating mechanism against glutamate. 

SOD enzymes are part of the antioxidant systems present in neurons, which help reduce damage caused by oxidative stress [18]. Decreasing SOD enzyme activity has been reported via glutamate treatment in agreement with our results [45,51]. In our experiments, SOD activity increased upon 3′-epilutein but not 3′-oxolutein; in parallel, 3′-epilutein was able to modify the SOD activity in glutamate treatments while 3′-oxolutein did not, referring to the different mechanisms of action of the two components. 

Oxidative stress can induce or contribute to the initiation of ferroptosis. Neurons are particularly susceptible to ferroptosis due to their high iron content, high metabolic rate, and limited capacity to replenish antioxidants. Oxidative stress can result in the accumulation of ROS, which can promote lipid peroxidation and initiate the ferroptotic cascade in neurons [22,52,53,54]. In addition, iron metabolism plays a significant role in the connection between oxidative stress and ferroptosis in neurons. Under normal physiological conditions, iron is tightly regulated and sequestered by proteins such as ferritin to prevent oxidative damage. The regulation of iron homeostasis includes the uptake and export of iron and iron storage [52].

TfR1 plays a crucial role in the cellular uptake of iron. In oxidative stress, the role of TfR1 in neurons is complex. During ferroptosis, the cell is unable to maintain the normal iron metabolism. TfR1 may contribute to the development of oxidative stress as iron enters neurons via TfR1. The higher expression of TfR1 results in more iron uptake and, as a result, further increases oxidative stress, which can damage neurons. On the other hand, TfR1 and ferroptosis also have an important function in the oxidative stress response in neurons; thus, high levels of labile iron are generated in the cell’s environment, with the increased uptake of iron may generate additional oxidative stress [26,55]. In our experiments, TfR1 expression was induced by glutamate treatment, while 3′-epilutein was able to downregulate the expression in the glutamate-treated cells. 

Ferritin in neurons has an essential function in iron storage and in protecting cells from oxidative stress. Ferritin has the capability to bind and store excess iron in cells, reducing unstable iron levels and preventing oxidative stress. Moreover, FTH expression may be regulated during ferroptosis in response to oxidative stress; thus, with increasing oxidative stress, FTH expression in neurons may upregulate, leading to increased ferritin formation [26,56]. In our results, glutamate-induced elevation in the relative mRNA expression of FTH while 3′-epilutein could downregulate the expression in each treatment. On the contrary, 3′-oxolutein could not modify the effects of glutamate in TfR1, nor in FTH expression. 

FPN is responsible for the transfer of iron from the cells. During ferroptosis, iron accumulates in cells, and ferroportin may be expressed at lower levels or present in an inactive form on the cell surface resulting in the inability to efficiently excrete excess iron and leading to the accumulation of iron in cells. In oxidative stress, the unstable iron may inhibit the function of FPN, which contribute to a further decrease in iron export [57,58]. Excessive glutamate stimulation may lead to a decrease in ferroportin expression and accumulation of intracellular iron in neurons which may cause further oxidative stress and contribute to neuronal damage or death [59]. In our experiments, glutamate treatments induced a slight decrease in the expression of FPN. Neither 3′-epilutein nor 3′-oxolutein altered FPN expression except 3′-epilutein in the presence of glutamate at 72 h. 

HO-1 plays a crucial role in the regulation of iron metabolism, ferroptosis, and oxidative stress, among others. The induction of HO-1 can increase HO-1 expression, and the release of iron may moderate intracellular iron levels and contribute to the inhibition of ferroptosis. The activation of HO-1 can be enhanced by oxidative stress, and as a result, an increase in HO-1 enhances the efficiency of the antioxidant defense system; the antioxidant effect of HO-1 may contribute to the reduction of oxidative stress and cell survival [20,60]. According to our results, HO-1 expression showed an increase upon glutamate treatment which was mitigated by 3′-epilutein and 3′-oxolutein. 

Ferroptosis is an iron-dependent type of cell death in which the combination of iron accumulation and oxidative stress leads to cell death. During lipid peroxidation, oxidized lipids, and breakdown products are formed, which can cause further detrimental effects on neurons along with serious implications for nervous system function [54]. ALOX5 and ALOX15 are major contributions to the generation of lipid peroxides; thus, their overexpression is related to diseases such as systemic sclerosis, certain types of cancers, and diabetes, moreover affecting inflammation and promoting lipid peroxides formation [61,62,63]. It has been recently revealed that ACSL4 plays a role in the development of lipid peroxidation and ferroptosis [64]. According to our results, both 3′-epilutein and 3′-oxolutein are able to downregulate the lipid peroxidizing genes upregulated by glutamate with a greater efficacy by 3′-epilutein. 

Since the accumulation of iron is a participant in ferroptosis and lipid peroxidation as well as oxidative stress, total iron measurements were performed to reveal the changes in intracellular iron levels of the SH-SY5Y cells. The content of iron was induced by glutamate treatment, while both 3′-epilutein and 3′-oxolutein could decrease it in the presence of glutamate. The components alone did not alter iron levels or other iron metabolism genes in our results, assuming their potential beneficial effects in oxidative stress. 

The regulation of lipid peroxidation and thiol compounds is tightly regulated in the maintenance of oxidative and antioxidative balance. Thiol compounds are a group of molecules containing sulfhydryl functional groups, such as glutathione, cysteine, and other thiol-containing proteins, which play a crucial role in cellular antioxidant defense systems. During lipid peroxidation, reactive lipid species can react with thiol compounds, leading to their depletion. As lipid peroxidation progresses and thiol compounds are consumed, the cellular antioxidant capacity might become compromised. This can lead to a vicious cycle, where lipid peroxidation-induced thiol depletion further exacerbates oxidative stress and lipid peroxidation, perpetuating cellular damage [30,31]. In our results, glutamate decreased the number of thiol compounds. Meanwhile, both 3′-epilutein and 3′-oxolutein have modified the effects of glutamate. Moreover, both carotenoids alone elevated the thiol compounds in SH-SY5Y cells suggesting their antioxidant-promoting effects. 

Oxidative stress can stimulate the inflammatory response in cells and increase the production of inflammatory cytokines. Inflammation is a fundamental component of the immune response; however, inflammatory cytokines, such as IL-1β, IL-6, IL-8, IFNγ, and TNFα, can stimulate the production of ROS, which can further induce oxidative damage. Neuroinflammation can be induced by oxidative stress, disease, or neuronal damage [65,66,67]. TNFα has been reported to increase glutamate-induced neurotoxicity in neuronal injury [68]. IL-8 has been shown to elevate neuroinflammation; thus, its dysregulation links to systemic inflammation [65]. Chronic levels of IL-6 have been demonstrated in neuronal dysfunction [69]. Elevated IL-8 and TNFα levels were published upon glutamate treatment in our previous article parallelly with the current work [51]. Both 3′-epilutein and 3′-oxolutein appeared as anti-inflammatory compounds in our experiments, mitigating the effects of glutamate-induced proinflammatory cytokine secretions in SH-SY5Y cells. Moreover, treatment with 3′-epilutein showed a greater downregulatory effect on IL-6 secretion compared to our previous experiments with lutein. 

## 4. Materials and Methods 

### 4.1. Preparation of 3′-Epilutein 

The preparation of 3′-epilutein was in accordance with our earlier work from lutein by acid-catalyzed epimerization [9]. Briefly, a solution of 200 mg of lutein (containing 6% of zeaxanthin) in tetrahydrofuran/water (THF/H2O) 1:1 (200 mL) was epimerized with 0.2% *w/v* aq. hydrogen chloride (HCl) solution (200 mL) at 25 °C for 48 h under N_2_ in the dark. The reaction was monitored by HPLC (Thermo Fisher Scientific Inc., Waltham, MA, USA). After 48 h, an aqueous solution of 5% sodium bicarbonate (NaHCO₃; Sigma-Aldrich, Budapest, Hungary) was added, and the crude product was partitioned into 300 mL of dimethyl ether. The aqueous layer was removed, then the organic layer was washed with water and dried over sodium sulfate (Na2SO4; Sigma-Aldrich, Budapest, Hungary). The mixture was separated by open column chromatography (CaCO_3_; benzene/hexane 2:3). The following fractions were obtained: Fraction 1, a mixture of (13Z)- and (13′Z)-isomers of lutein; Fraction 2, a mixture of (9Z)-and (9′Z)-lutein, together with 3′-epilutein as a main component; Fraction 3, (Z)-isomers of 3′-epilutein; Fraction 4, a mixture of lutein and zeaxanthin; Fraction 5, a mixture of anhydroluteins. Repeated OCC of Fraction 2 (CaCO_3_, acetone/hexane 4:96) resulted in Fraction 21 ((9Z)- and (9′Z)-lutein) and Fraction 22 (3′-epilutein). After recrystallization from benzene/hexane (1:5), 24 mg of 3′-epilutein was obtained.

### 4.2. Preparation of 3′-Oxolutein

The preparation of 3′-oxolutein was carried out similarly to our earlier work. Briefly, 500 mg of natural lutein extract (contains 6% of zeaxanthin) was dissolved in 1.2 L of acetone (Molar Chemicals Ltd., Halásztelek, Hungary) and 8.4 g of activated manganese dioxide (MnO_2_; Sigma-Aldrich, Budapest, Hungary) was added. The mixture was vigorously stirred for at least 12 h. When thin-layer chromatography (TLC) showed almost complete conversion, the mixture was filtered through Celite (Celite^®^ 545; Cat. No. 68855-54-9; Merck Life Science Kft., Budapest, Hungary), and the filtrate was concentrated. The crude product was purified on a silica gel 60 (Molar Chemicals Ltd., Halásztelek, Hungary) column using hexane:acetone 85:15 as eluent. Crystallization from toluene/hexane delivered 282 mg of 3′-oxolutein with 99% HPLC purity (recorded at 450 nm) (Dionex; Thermo Fisher Scientific Inc., Waltham, MA, USA). The data acquisition was performed by Chromeleon 6.70 Software. The HPLC separation was carried out on an end-capped C30 column (250 × 4.6 mm i.d.; YMC C30, 3 µm). Eluents: (A) MeOH:MTBE:H2O(81:15:4); (B) MeOH:MTBE:H_2_O = 6:90:4. The chromatography was performed in a linear gradient from 100% A eluent to 50% B mixture in 45 min, with 1 cm^3^/min flow. All spectroscopic data were in accordance with the literature [13]. 

### 4.3. Cell Culture and Treatments

SH-SY5Y human neuroblastoma cell line (ATCC; CRL-2266) was used for the cell culture experiments. The cells were cultured and maintained in Dulbecco’s Modified Eagle Medium/Nutrient Mixture-F12 (DMEM-F12; Cat. No. L0093, Biowest, Nuaillé, France) medium supplemented with 10% Fetal Bovine Serum (FBS; Cat. No. S1810, Biowest, Nuaillé, France) and 1% Penicillin/Streptomycin (P/S; Lonza Ltd., Basel, Switzerland) under a humidified atmosphere containing 5% CO_2_ at 37 °C. In the experiments, the cells were seeded onto 96-well or 6-well plates or into 25 cm^2^ tissue culture flasks in an antibiotic-free complete media and rested for 24 h before the treatments. To induce oxidative stress, the cells were treated with 1 mM or 5 mM of glutamate (Merck Life Science Kft., Budapest, Hungary) according to our previous literature [51]. The 3′-epilutein or 3′-oxolutein was dissolved in dimethyl sulfoxide (DMSO; Sigma-Aldrich, Budapest, Hungary) as a carrier. The cells were treated with 10 ng/µL of 3′-epilutein or 3′-oxolutein to follow the effects of these carotenoids. The untreated cells served as an absolute control; the DMSO-treated cells were used as a control for the glutamate and 3′-epilutein or 3′-oxolutein treatments. The following abbreviations were used in this work: C—control; G1—1 mM of glutamate; G5—5 mM of glutamate; E—3′-epilutein; O—3′-oxolutein; EG1—3′-epilutein with 1 mM of glutamate; EG5—3′-epilutein with 5 mM of glutamate; OG1—3′-oxolutein with 1 mM of glutamate; OG5—3′-oxolutein with 5 mM of glutamate. 

### 4.4. Viability Measurements

The viability assay measurements were performed by using a Resazurin based In Vitro Toxicology Assay Kit (Sigma-Aldrich, Budapest, Hungary). The cells were maintained as described above. For the viability measurements, 5 × 10^5^ cells/well were treated on 96-well plates in 100 µL of cell culture media. The following concentrations of 3′-epilutein and 3′-oxolutein were used in the preliminary experiments: 2.5 ng/µL, 5 ng/µL, 7.5 ng/µL, and 10 ng/µL and incubated for 24 h, 48 h, and 72 h. Next, viability measurements were carried out after treatments with 10 ng/µL of 3′-epilutein or 3′-oxolutein with 1 mM or 5 mM of glutamate. Each experiment was repeated in quadruplicate. The results of the viability measurements can be seen in the Appendix A. 

### 4.5. Reactive Oxygen Species Detection 

A fluorometric Intracellular ROS kit (Deep Red) (Cat. No. MAK142; Sigma-Aldrich, Budapest, Hungary) was used to measure the oxidative stress in the cells. The experiments were carried out on a 96-well plate according to the manufacturer’s instructions. For the measurements, 5 × 10^5^ cells/well were treated with 10 ng/µL of 3′-epilutein or 3′-oxolutein and with 1 mM or 5 mM of glutamate or together 3′-epilutein or 3′-oxolutein first, followed by glutamate treatment. Briefly, the cells were rested for 24 h after seeding, then treated with 3′-epilutein or 3′-oxolutein or glutamate to generate ROS production in the cells. The treated plates were incubated for 10, 20, and 30 min in 5% CO_2_ at 37 °C, then the cells were stained with 100 µL/well of ROS deep red dye solution and were incubated for 30 min. The absorbance was detected with a bottom read mode by EnSpire Multimode microplate reader (PerkinElmer, Rodgau, Germany) at 650/675 nm excitation/emission wavelengths. The alteration in ROS was determined as a percentage of control. 

### 4.6. Total Antioxidant Capacity Measurements

Total Antioxidant Capacity Assay Kit (Cat. No. MAK187; Sigma-Aldrich, Budapest, Hungary) measurements were carried out to follow the oxidative stress-induced damage in the cells. The experiments were performed on a 96-well tissue culture plate according to the manufacturer’s protocol. The kit applies Cu^2+^ as a sensor for the antioxidant capacity. Trolox, a vitamin E analog, is used as standard in the following concentrations: 0, 4, 8, 12, 16, and 20 nmol/well. The kit allows detection of the protein antioxidants, also able to measure the small molecule antioxidants by using the kit’s protein mask in a 1:1 ratio with the samples. The procedure was performed upon the instruction of the manufacturers. Briefly, 100 µL of the samples and 100 µL of the diluted standards were used for the measurements, then 100 µL of the working reagent was added to each well. The reagents were mixed by using an orbital shaker (BioSan, Riga, Latvia), and the plate was incubated for 90 min protected from light. Each experiment was repeated three times and carried out at room temperature in duplicate. The reduced Cu^+^ chelates with the provided colorimetric probe on absorbance at 570 nm, which was detected by using a MultiSkan GO spectrophotometer (Thermo Fisher Scientific Inc., Waltham, MA, USA). 

### 4.7. Thiol Measurements 

The fluorometric Thiol Quantitation Kit (Cat. No. MAK151; Sigma-Aldrich, Budapest, Hungary) was used to quantify the thiol content of small molecules. The experiments were carried out on a 96-well plate, with each sample and standard in duplicate. The measurements were performed according to the manufacturer’s instructions and repeated three times. Briefly, GSH standards were diluted with Assay Buffer to generate 0 µM, 0.14 µM, 0.04 µM, 0.12 µM, 0.37 µM, 1.1 µM, 3.3 µM and 10 µM standards for the measurements. First, 50 µL of the standards and samples were added to the wells. Next, 50 µL of Thiol Detection Reagent containing Master Reaction Mix was pipetted onto the plate, mixed well then incubated for 60 min at room temperature, protected from light. The fluorescence intensity was detected at 490/535 nm of excitation and emission wavelength by EnSpire Multimode Plate Reader (Perkin Elmer, Waltham, MA, USA). The results were calculated with the values of the standard curve of GSH standards. 

### 4.8. SOD Activity Measurements

The SOD activity measurements were performed by using SOD Activity Assay Kit (Cat. No. CS0009; Sigma-Aldrich, Budapest, Hungary). The experiments were carried out on a 96-well plate, all standards, and samples in triplicate. The procedure was performed according to the manufacturer’s instructions. Briefly, 1 × 10^6^/flask of treated cells were collected after incubation and were lysed in an ice-cold lysis buffer (0.1 M Trizma-HCl, containing 0.5% Triton X-100, 5 mM mercaptoethanol and protease inhibitors) by using an ultrasound sonicator. The homogenates were centrifugated at 4 °C on 14,000× *g* for 5 min then supernatants were transferred into new tubes for the experiments. Standards were prepared by using the SOD enzyme containing working solutions to generate 0, 0.3, 0.6, 0.9, 1.5, 3, and 6 units/mL of standards and diluted with the Dilution Buffer in the plate wells, each standard in 20 µL of total volume. Next, 20 µL of samples were added to the wells, then 160 µL of WST working solution was pipetted to the standards, samples, controls, and blank wells. The plate was incubated for 30 min at room temperature. The absorbance was detected at 450 nm by using a MultiSkan GO spectrophotometer (Thermo Scientific Inc., Waltham, MA, USA). The results were calculated upon the provided Excel-based calculation sheet and are presented as the SOD activity in units/mL. 

### 4.9. Real-Time PCR Experiments

SYBR Green-based real-time PCR experiments were performed to follow the alteration in gene expression in SH-SY5Y cells. The cells were treated the same way as described above, in a 6-well tissue culture plate with 3 × 10^5^ cells/well. After incubation, the cells were collected by trypsinization. Aurum™ Total RNA Mini Kit (Bio-Rad Inc., Hercules, CA, USA) was used for the isolation of total RNA according to the instructions of manufacturers. Then complementary DNA was synthesized from 200 ng of total RNA by using the iScript™ Select cDNA Synthesis Kit (Bio-Rad Inc., Hercules, CA, USA) in accordance with the manufacturer’s protocol. The determination of gene expression was performed by a CFX96 Real-Time System (Bio-Rad Inc., Hercules, CA, USA) using iTaq™ Universal SYBR^®^ Green Supermix (Bio-Rad Inc., Hercules, CA, USA) in 20 µL of total reaction volume. The melting curves were generated after each PCR run to ensure that the single specific product was amplified. The relative quantification was calculated by the Livak (∆∆Ct) method using the Bio-Rad CFX Maestro 3.1 software (Bio-Rad Inc., Hercules, CA, USA). The expression level of the target gene was compared to the level of β-actin in all samples. Then the relative expression rates were compared between the untreated and treated samples. The relative expression of the controls was regarded as 1 and is not indicated on the column diagrams. The mRNA expression of the treated cells was compared to the appropriate controls in each experiment. The primer sequences used in this work are represented in Table 1. 

### 4.10. ELISA Measurements

SH-SY5Y cells were treated the same way as described above. After the treatments, the culture media of the control and treated cells were collected and stored at −80 °C until the ELISA measurements. Human IL-6, IL-8, and TNFα ELISA Kits (Invitrogen, Thermo Fisher Scientific Inc., Waltham, MA, USA) were used to measure the secreted cytokine levels of the samples according to the instructions of the manufacturer. MultiSkan GO spectrophotometer (Thermo Fisher Scientific Inc., Waltham, MA, USA) was used for the detection of the absorbance at 450 nm. The intensity of the signal is directly proportional to the concentration of the target cytokines present in the original samples. The concentrations of the proinflammatory cytokines were expressed and presented as pg/mL. 

### 4.11. Total Iron Determinations 

A ferrozine-based colorimetric assay was used to determine the total iron levels of the cells in a procedure described by Riemer et al. [70]. For the iron measurements, 1 × 10^6^ cells/flask were treated as described above and collected after the incubation. The cells were lysed by using an orbital shaker (BioSan, Riga, Latvia) in 50 mM sodium hydroxide (NaOH; Sigma-Aldrich, Budapest, Hungary) for 2 h at 25 °C. Then the lysed cells were mixed and incubated with an iron-releasing reagent (1.4 M hydrogen chloride (HCl) (4.5% (wt/vol)) and potassium permanganate (KMnO_4_; Sigma-Aldrich, Budapest, Hungary) in distilled water (dH2O) for 2 h at 60 °C. The samples were cooled, and the released iron was chelated by an iron detection reagent (6.5 mM ferrozine, 6.5 mM neocuproine, 2.5 M ammonium acetate, and 1 M ascorbic acid (Sigma-Aldrich, Budapest, Hungary)) and incubated for 30 min at 25 °C. MultiSkan GO spectrophotometer (Thermo Fisher Scientific Inc., Waltham, MA, USA) was used for the detection of the absorbance at 550 nm. The concentration of iron was calculated by ferric chloride (FeCl_3_; 0–300 µM) standard curve that was treated the same as the samples. The protein concentrations of the samples were detected by DC Protein Assay Kit (Bio-Rad Inc., Hercules, CA, USA). The level of iron was normalized against the concentration of protein and expressed as µM iron/mg protein. 

### 4.12. Statistical Analysis 

The data presented in this work are representative of at least three independent experiments. The n corresponds to the number of independent experiments. The viability and ROS measurements were performed in quadruplicate in all independent experiments. Thiol quantifications, TAC, ELISA and Total iron measurements, SOD assays, and real-time PCR experiments were carried out in triplicates in each independent experiment. For the statistical analysis, SPSS 24.0 software (IBM Corporation, Armonk, NY, USA) was used. The statistical significance was determined by using ANOVA analyses with Tukey post hoc tests comparing the 3′-epilutein or 3′-oxolutein treated groups or the glutamate treatments at 24 h, 48 h, and 72 h to the controls and the 3′-epilutein or 3′-oxolutein treated groups compared to their mutual treatments with glutamate. The presented data are the mean values with the error bars corresponding to standard deviation (±SD). The significance was set at a *p*-value less than 0.05 (*p* < 0.05). Asterisk represents *p* < 0.05 between 3′-epilutein or 3′-oxolutein or glutamate treatment and control cell; cross indicates *p* < 0.05 between 3′-epilutein or 3′-oxolutein treated groups compared to their mutual treatments with glutamate.

## 5. Conclusions

We first revealed that 3′-epilutein and 3′-oxolutein, the natural metabolites of lutein in the human body, were able to provide antioxidant and anti-inflammatory protection against glutamate-induced oxidative stress in SH-SY5Y cells; thus, 3′-epilutein appeared to supply a greater impact as a protective agent. Through their antioxidant activity, these carotenoids can protect nerve cells from harmful effects, reduce oxidative stress and slow down disease progression. Neurodegenerative diseases are often associated with neuroinflammation in the nervous system. Given the possibility, for further studies, combined use of 3′-epilutein and lutein may have a synergistic effect, such as they may have a mutually reinforcing effect in the treatment of neurodegenerative diseases. Working together, the compounds may improve intercellular communication, reduce oxidative stress and inflammation, and protect nerve cells. It is important to note, however, that the effects of 3′-epilutein and lutein are still an active area of research, and further clinical trials are needed. 

## Figures and Tables

**Figure 1 ijms-24-12008-f001:**
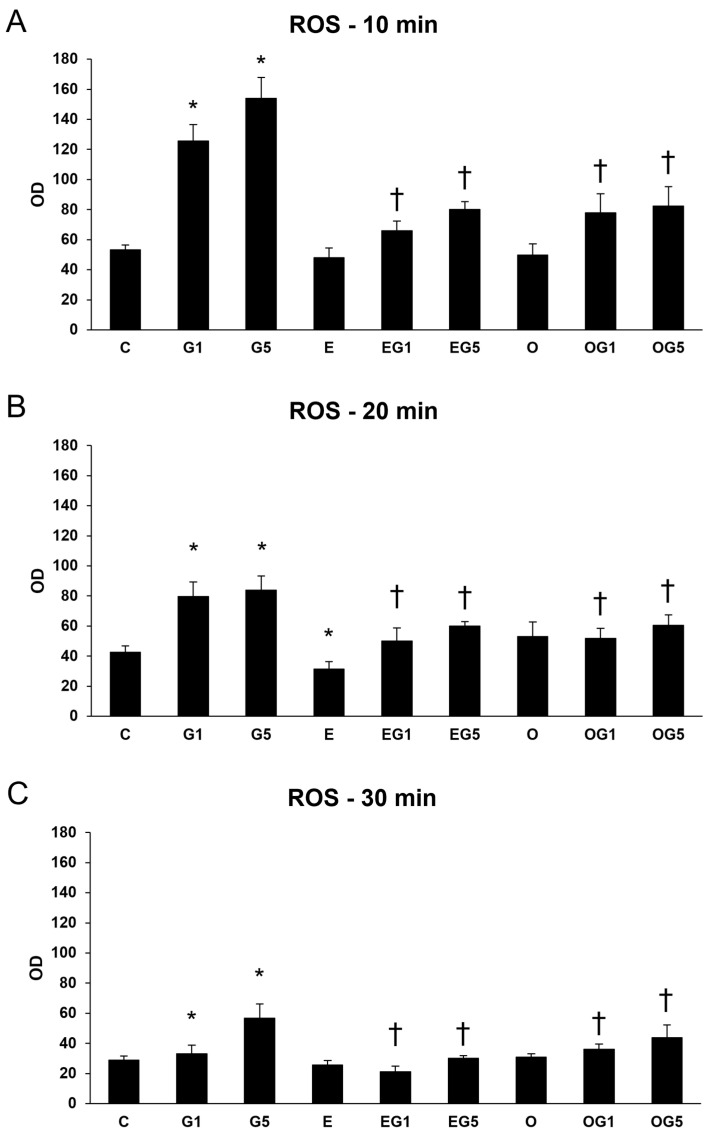
Reactive oxygen species (ROS) detection after 3′-epilutein and 3′-oxolutein and glutamate treatments in SH-SY5Y cells at 10 min (**A**), at 20 min (**B**), and at 30 min (**C**). The changes in ROS were determined as a percentage of control. The bars represent the mean values (±SD) of three independent experiments (n = 3), each performed in quadruplicate, and are presented relative to their control cells. For better transparency, the measured ROS of DMSO controls is not presented. The asterisk signs the statistical significance of glutamate treatments compared to control; cross marks the statistical significance of combined (e.g., 3′-epilutein with glutamate) treatments compared to glutamate treatments at 10 min, 20 min, or 30 min (*p* < 0.05). Abbreviations: C—control; G1—1 mM glutamate; G5—5 mM glutamate; E—3′-epilutein; O—3′-oxolutein; EG1—3′-epilutein + 1 mM glutamate; EG5—3′-epilutein + 5 mM glutamate; OG1—3′-oxolutein + 1 mM glutamate; OG5—3′-oxolutein + 5 mM glutamate; OD—optical density.

**Figure 2 ijms-24-12008-f002:**
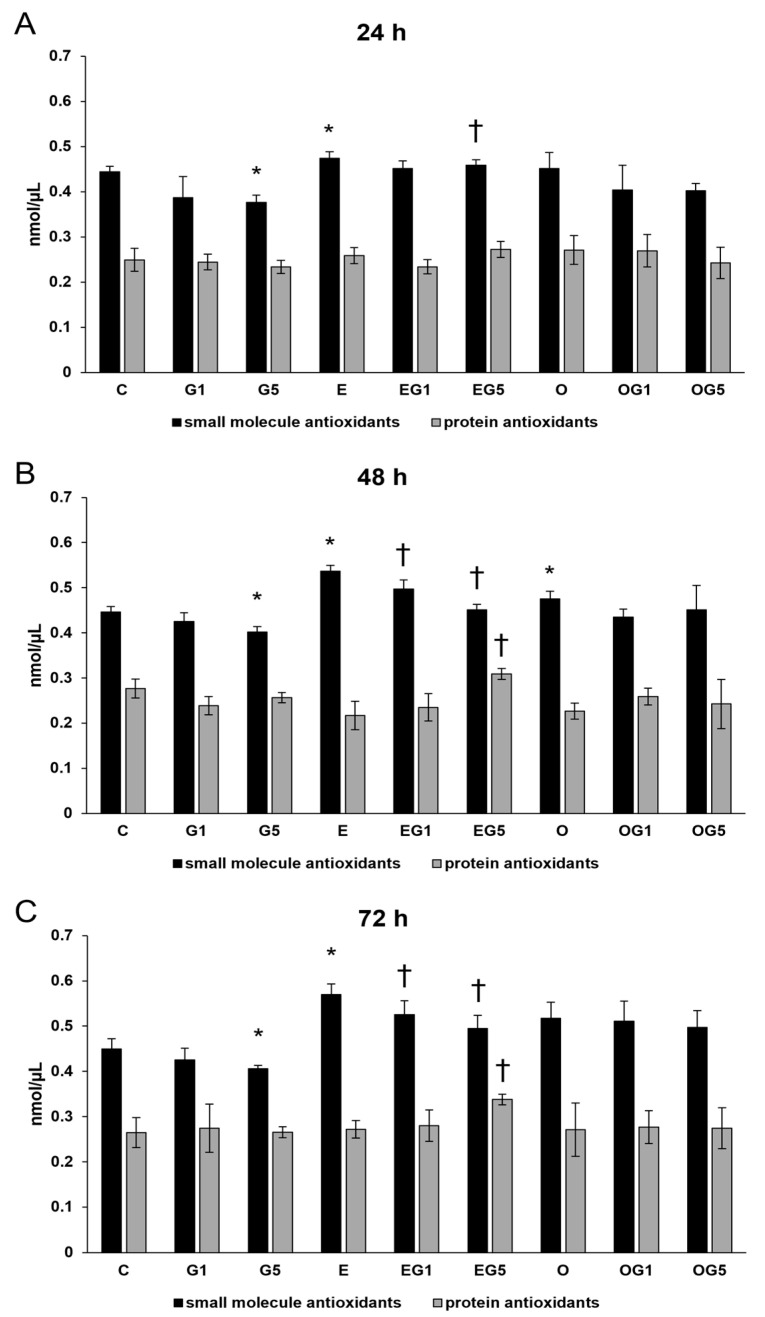
Total antioxidant capacity (TAC) determinations after 3′-epilutein and 3′-oxolutein and glutamate treatments in SH-SY5Y cells at 24 h (**A**), at 48 h (**B**), and at 72 h (**C**). The columns represent the mean values in nmol/µL, and error bars indicate the standard deviation (SD) of three independent experiments (n = 3), each performed in quadruplicate, and are presented relative to their own control cells. Asterisk indicates the statistical significance of glutamate-treated cells compared to controls at 24 h, 48 h, and 72 h; cross marks the statistical significance of combined (e.g., 3′-epilutein with glutamate) treatments compared to glutamate treatments at 24 h, 48 h, and 72 h; the level of significance was *p* < 0.05. Abbreviations: C—control; G1—1 mM glutamate; G5—5 mM glutamate; E—3′-epilutein; O—3′-oxolutein; EG1—3′-epilutein + 1 mM glutamate; EG5—3′-epilutein + 5 mM glutamate; OG1—3′-oxolutein + 1 mM glutamate; OG5—3′-oxolutein + 5 mM glutamate.

**Figure 3 ijms-24-12008-f003:**
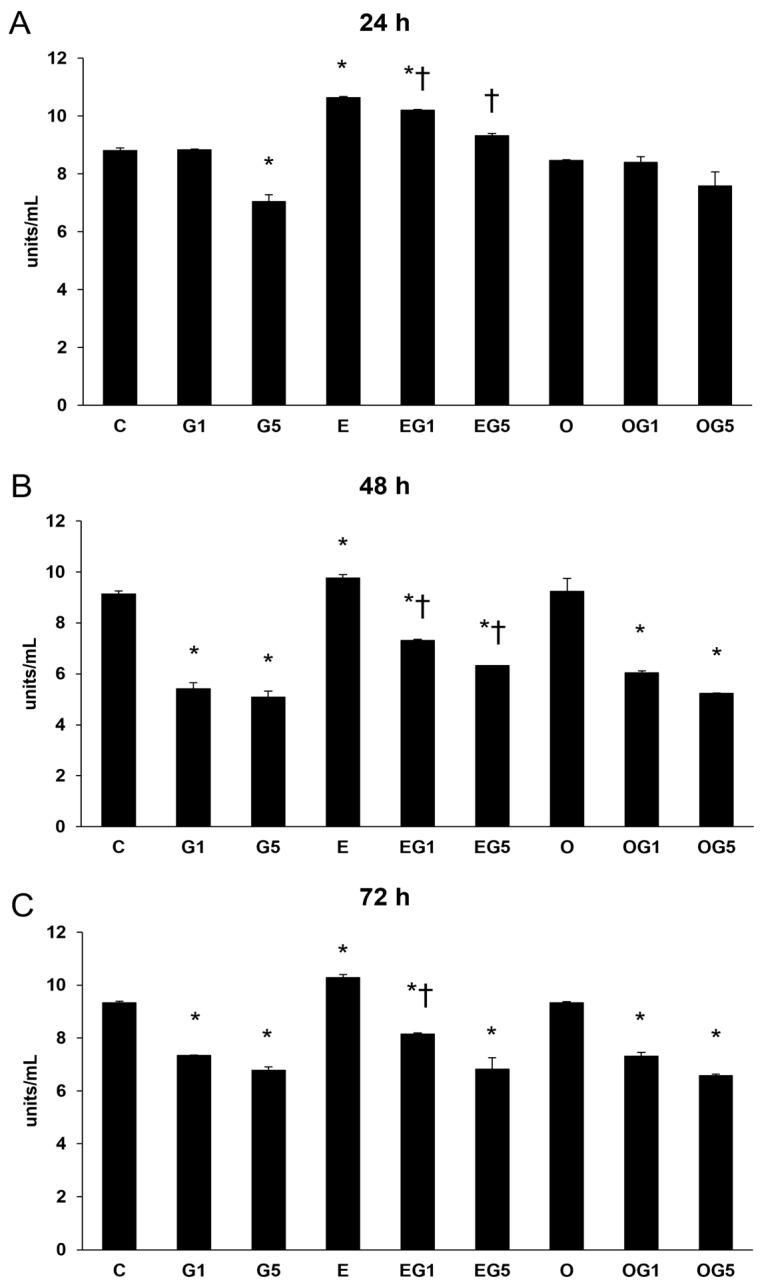
Determination of the effects of 3′-epilutein and 3′-oxolutein on SOD activity of the SH-SY5Y cells at 24 h (**A**), at 48 h (**B**), and at 72 h (**C**). The result was expressed as U/mL. The bars represent the mean values (±SD) of three independent experiments (n = 3) performed in triplicate and are presented relative to their control cells. Asterisk marks the statistical significance of glutamate treatments compared to control, cross represents the statistical significance of combined (e.g., 3′-epilutein + glutamate) treatments compared to the glutamate treatments at 24 h, 48 h, and 72 h (*p* < 0.05). Abbreviations: C—control; G1—1 mM glutamate; G5—5 mM glutamate; E—3′-epilutein; O—3′-oxolutein; EG1—3′-epilutein + 1 mM glutamate; EG5—3′-epilutein + 5 mM glutamate; OG1—3′-oxolutein + 1 mM glutamate; OG5—3′-oxolutein + 5 mM glutamate.

**Figure 4 ijms-24-12008-f004:**
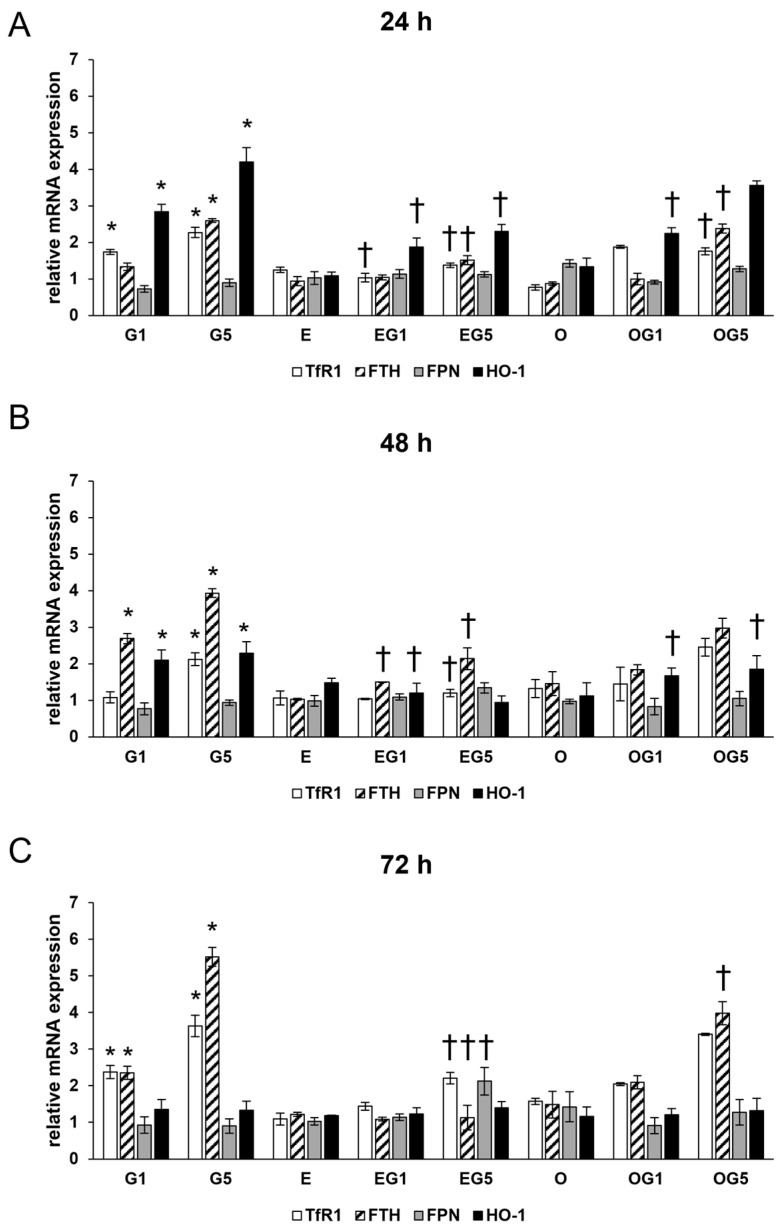
Relative mRNA expression of TfR1, FTH, FPN, and HO-1 in 3′-epilutein and 3′-oxolutein and glutamate-treated SH-SY5Y cells at 24 h (**A**), at 48 h (**B**), and at 72 h (**C**). The relative expression of controls was regarded as 1 and is not indicated on the column diagrams. The bars represent the mean values (±SD) of three independent experiments (n = 3) performed in triplicate and are presented relative to their control cells. Asterisk marks the statistical significance of glutamate treatments compared to control, cross represents the statistical significance of combined (e.g., 3′-epilutein + glutamate) treatments compared to the glutamate treatments at 24 h, 48 h, and 72 h (*p* < 0.05). Abbreviations: G1—1 mM glutamate; G5—5 mM glutamate; E—3′-epilutein; O—3′-oxolutein; EG1—3′-epilutein + 1 mM glutamate; EG5—3′-epilutein + 5 mM glutamate; OG1—3′-oxolutein + 1 mM glutamate; OG5—3′-oxolutein + 5 mM glutamate.

**Figure 5 ijms-24-12008-f005:**
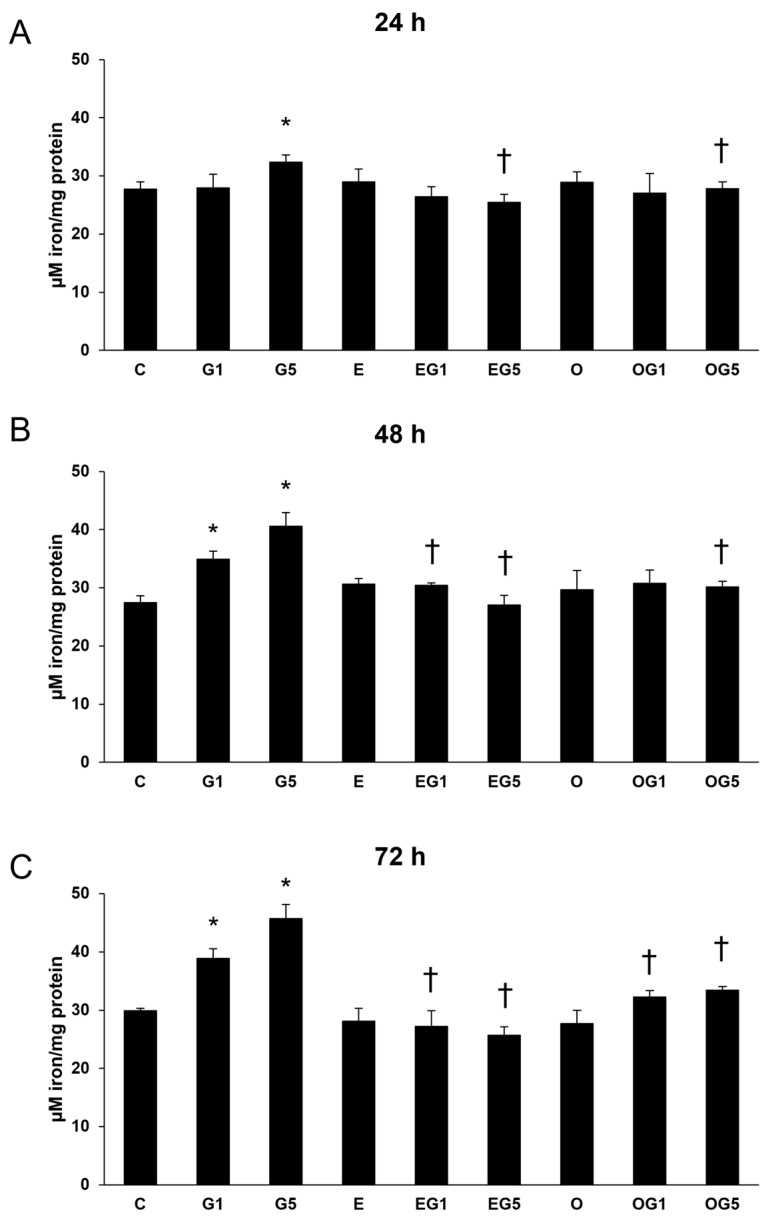
Determinations of iron content in 3′-epilutein and 3′-oxolutein and glutamate-treated SH-SY5Y cells at 24 h (**A**), at 48 h (**B**), and at 72 h (**C**). The results of iron levels were normalized against the protein concentration and were expressed as µM iron/mg protein. The bars represent the mean values (±SD) of three independent experiments (n = 3) performed in triplicate and are presented relative to their control cells. Asterisk represents the statistical significance of glutamate treatments compared to control; cross signs the statistical significance of combined (e.g., 3′-epilutein + glutamate) treatments compared to the glutamate treatments at 24 h, 48 h, and 72 h (*p* < 0.05). Abbreviations: C—control; G1—1 mM glutamate; G5—5 mM glutamate; E—3′-epilutein; O—3′-oxolutein; EG1—3′-epilutein + 1 mM glutamate; EG5—3′-epilutein + 5 mM glutamate; OG1—3′-oxolutein + 1 mM glutamate; OG5—3′-oxolutein + 5 mM glutamate.

**Figure 6 ijms-24-12008-f006:**
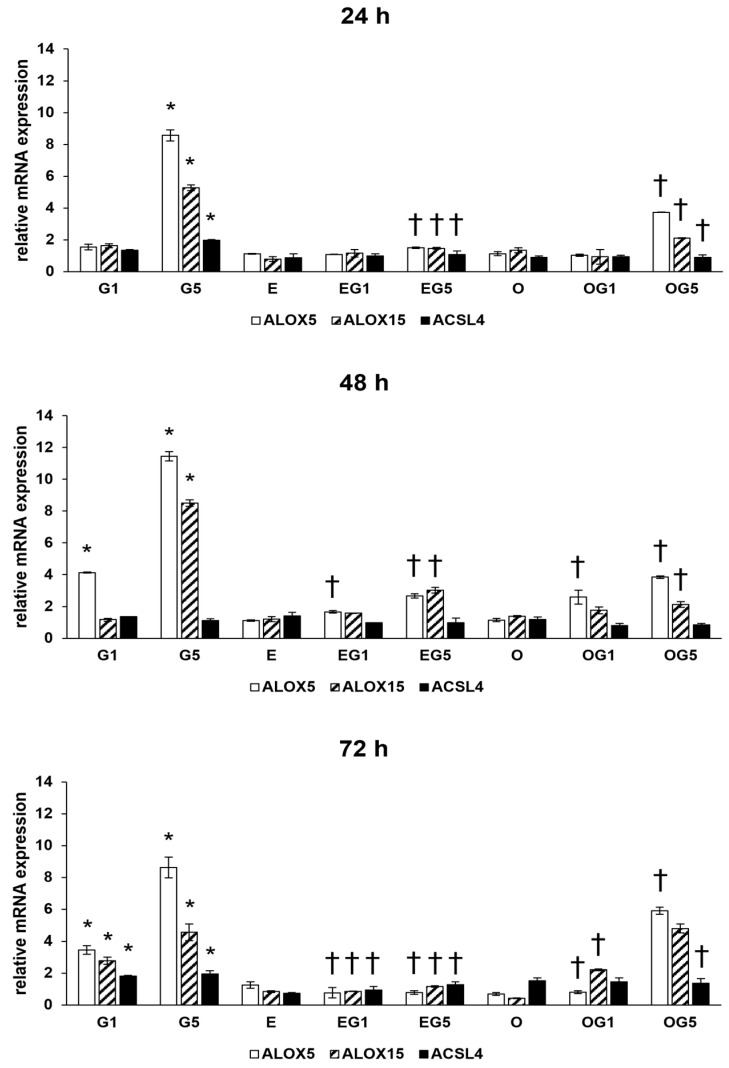
Relative mRNA expression of ALOX5, ALOX15, and ACSL4 in 3′-epilutein and 3′-oxolutein and glutamate-treated SH-SY5Y cells. The relative expression of controls was regarded as 1 and is not indicated on the column diagrams. The bars represent the mean values (±SD) of three independent experiments (n = 3) performed in triplicate and are presented relative to their control cells. Asterisk signs the statistical significance of glutamate treatments compared to control; cross marks the statistical significance of combined (e.g., 3′-epilutein + glutamate) treatments compared to the glutamate treatments at 24 h, 48 h, and 72 h (*p* < 0.05). Abbreviations: G1—1 mM glutamate; G5—5 mM glutamate; E—3′-epilutein; O—3′-oxolutein; EG1—3′-epilutein + 1 mM glutamate; EG5—3′-epilutein + 5 mM glutamate; OG1—3′-oxolutein + 1 mM glutamate; OG5—3′-oxolutein + 5 mM glutamate.

**Figure 7 ijms-24-12008-f007:**
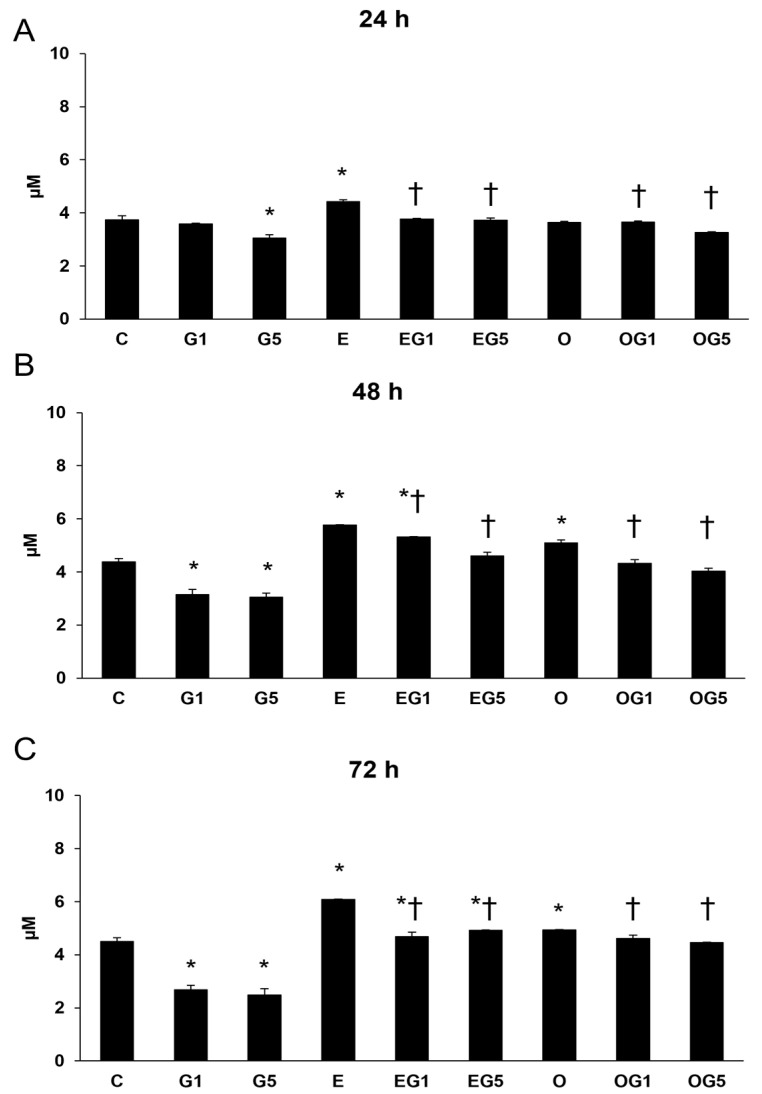
Thiol determinations after 3′-epilutein and 3′-oxolutein and glutamate treatments in SH-SY5Y cells at 24 h (**A**), at 48 h (**B**), and at 72 h (**C**). The columns represent the mean values in µM, and error bars indicate the standard deviation (±SD) of three independent experiments (n = 3), each performed in quadruplicate, and are presented relative to own control cells. Asterisk shows the statistical significance of glutamate-treated cells compared to controls at 24 h, 48 h, and 72 h; cross signs statistical significance of combined (e.g., 3′-epilutein with glutamate) treatments compared to glutamate treatments at 24 h, 48 h, and 72 h; the level of significance was *p* < 0.05. Abbreviations: C—control; G1—1 mM glutamate; G5—5 mM glutamate; E—3′-epilutein; O—3′-oxolutein; EG1—3′-epilutein + 1 mM glutamate; EG5—3′-epilutein + 5 mM glutamate; OG1—3′-oxolutein + 1 mM glutamate; OG5—3′-oxolutein + 5 mM glutamate.

**Figure 8 ijms-24-12008-f008:**
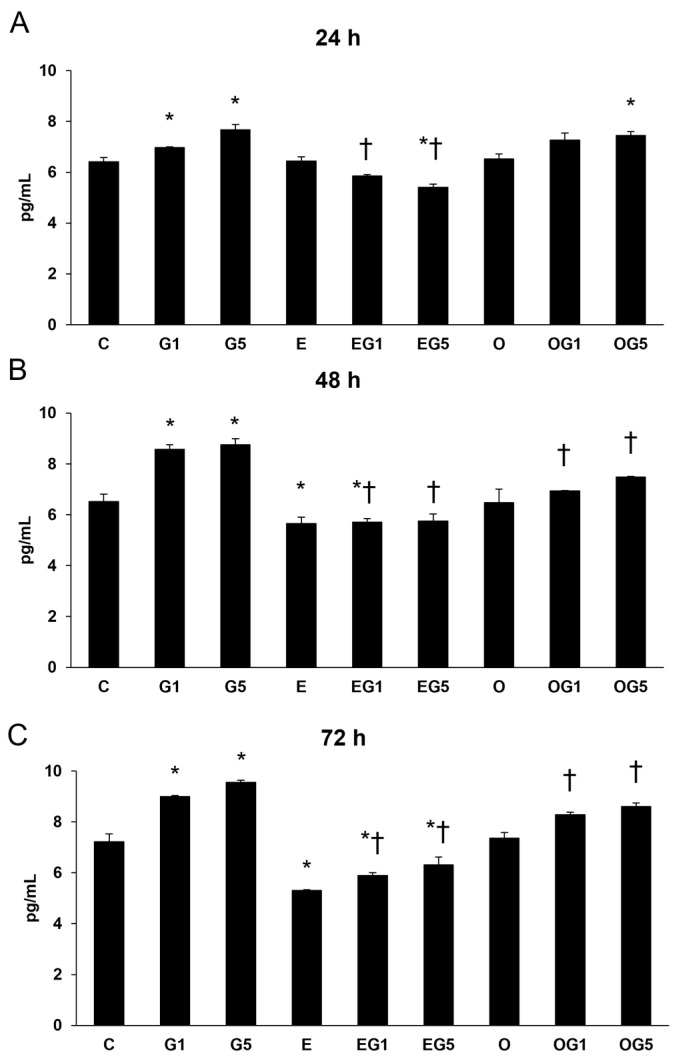
Determination of proinflammatory IL-6 secretion after 3′-epilutein, 3′-oxolutein, and glutamate treatments of the SH-SY5Y cells at 24 h (**A**), at 48 h (**B**), and at 72 h (**C**). The columns represent the mean values expressed in pg/mL concentration, and error bars indicate the standard deviation (±SD) of three independent experiments (n = 3), each performed in quadruplicate, and are presented relative to own control cells. Asterisk shows the statistical significance of glutamate-treated cells compared to controls at 24 h, 48 h, and 72 h; cross signs statistical significance of combined (e.g., 3′-epilutein with glutamate) treatments compared to glutamate treatments at 24 h, 48 h, and 72 h; the level of significance was *p* < 0.05. Abbreviations: C—control; G1—1 mM glutamate; G5—5 mM glutamate; E—3′-epilutein; O—3′-oxolutein; EG1—3′-epilutein + 1 mM glutamate; EG5—3′-epilutein + 5 mM glutamate; OG1—3′-oxolutein + 1 mM glutamate; OG5—3′-oxolutein + 5 mM glutamate.

**Figure 9 ijms-24-12008-f009:**
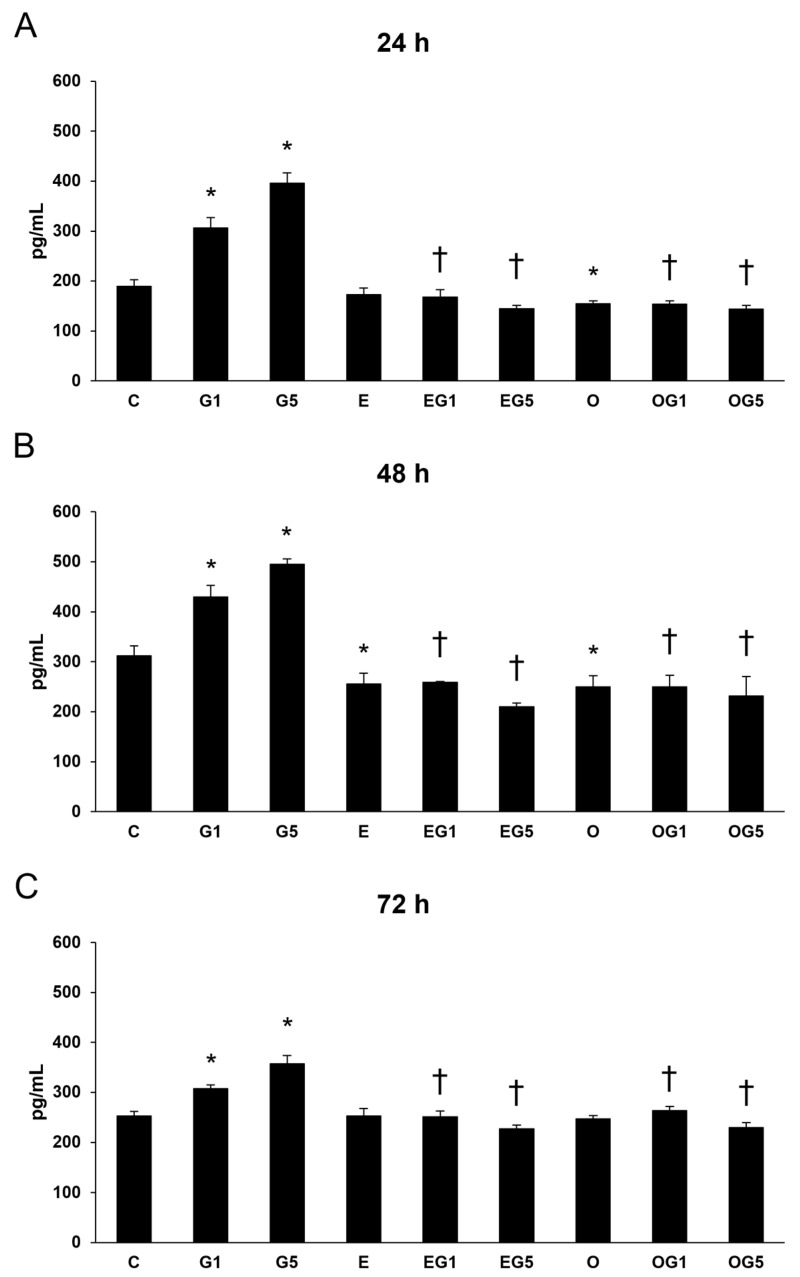
Determination of proinflammatory IL-8 secretion after 3′-epilutein, 3′-oxolutein, and glutamate treatments of the SH-SY5Y cells at 24 h (**A**), at 48 h (**B**), and at 72 h (**C**). The columns represent the mean values expressed in pg/mL concentration. Error bars indicate the standard deviation (±SD) of three independent experiments (n = 3), each performed in quadruplicate, and are presented relative to own control cells. Asterisk marks the statistical significance of glutamate-treated cells compared to controls at 24 h, 48 h, and 72 h; cross signs statistical significance of combined (e.g., 3′-epilutein with glutamate) treatments compared to glutamate treatments at 24 h, 48 h, and 72 h; the level of significance was *p* < 0.05. Abbreviations: C—control; G1—1 mM glutamate; G5—5 mM glutamate; E—3′-epilutein; O—3′-oxolutein; EG1—3′-epilutein + 1 mM glutamate; EG5—3′-epilutein + 5 mM glutamate; OG1—3′-oxolutein + 1 mM glutamate; OG5—3′-oxolutein + 5 mM glutamate.

**Figure 10 ijms-24-12008-f010:**
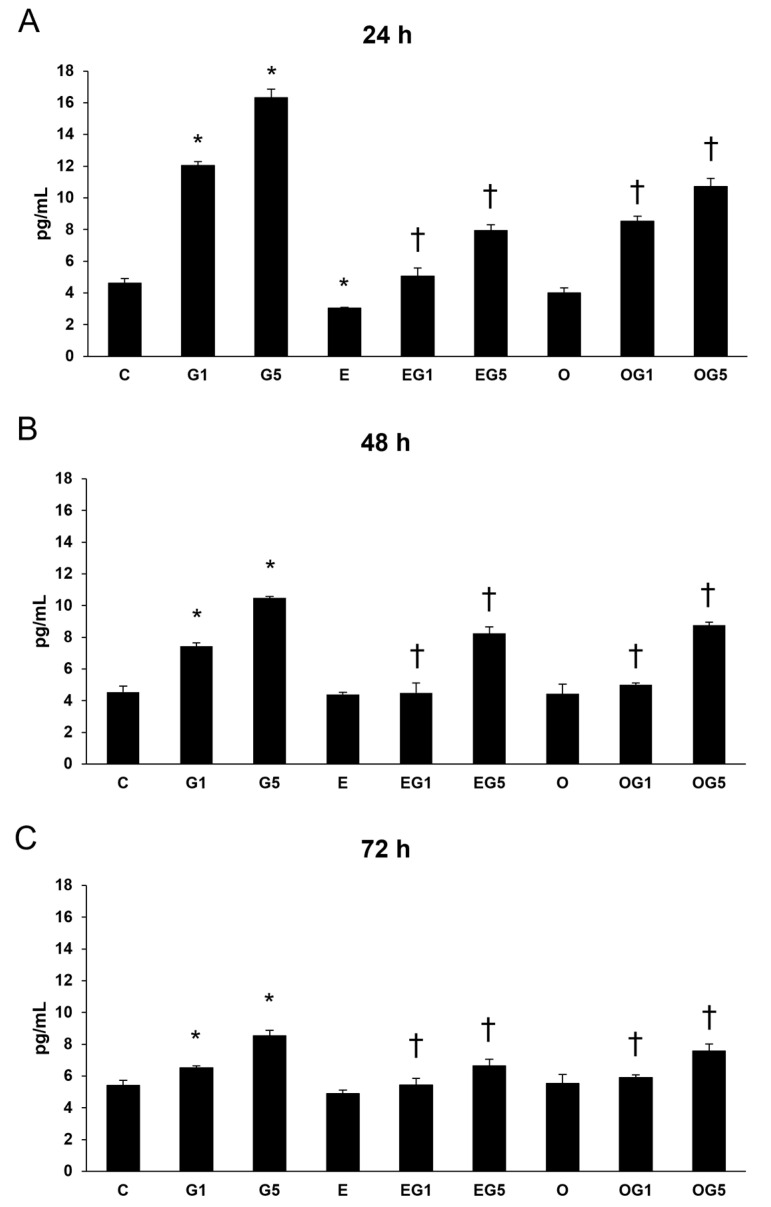
Determination of TNFα secretion after 3′-epilutein, 3′-oxolutein, and glutamate treatments of the SH-SY5Y cells at 24 h (**A**), at 48 h (**B**), and at 72 h (**C**). The columns represent the mean values expressed in pg/mL concentration, and error bars indicate the standard deviation (±SD) of three independent experiments (n = 3), each performed in quadruplicate, and are presented relative to own control cells. Asterisk signs statistical significance of glutamate-treated cells compared to controls at 24 h, 48 h, and 72 h, cross represents the statistical significance of combined (e.g., 3′-epilutein with glutamate) treatments compared to glutamate treatments at 24 h, 48 h, and 72 h; the level of significance was *p* < 0.05. Abbreviations: C—control; G1—1 mM glutamate; G5—5 mM glutamate; E—3′-epilutein; O—3′-oxolutein; EG1—3′-epilutein + 1 mM glutamate; EG5—3′-epilutein + 5 mM glutamate; OG1—3′-oxolutein + 1 mM glutamate; OG5—3′-oxolutein + 5 mM glutamate.

**Table 1 ijms-24-12008-t001:** Real-time PCR gene primer sequence list.

Target Gene	Sequence 5′ → 3
β-actin forward	AGAAAATCTGGCACCACACC
β-actin reverse	GGGGTGTTGAAGGTGTCAAA
ACSL4 forward	TCTTGCTTTACCTATGGCTG
ACSL4 reverse	CAGTACAGTCTCCTTTGCTT
ALOX5 forward	CGCGGTGGATTCATACG
ALOX5 reverse	GTCTTCAGCGTGATGTACT
ALOX15 forward	GAGGAGGAGTATTTTTCGGG
ALOX15 reverse	AATTTCCTTATCCAGGGCAG
FPN forward	AAAGGAGGCTGTTTCCATAG
FPN reverse	TTCCTTCTCTACCTTGGTCA
FTH forward	GAGGTGGCCGAATCTTCCTTC
FTH reverse	TCAGTGGCCAGTTTGTGCAG
HO-1 forward	ACCCATGACACCAAGGACCA
HO-1 reverse	ATGCCTGCATTCACATGGCA
TfR1 forward	CATGTGGAGATGAAACTTGC
TfR1 reverse	TCCCATAGCAGATACTTCCA

## Data Availability

Data are contained within the article and Appendix A.

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
