# Peer review of "Protective Effects of 3′-Epilutein and 3′-Oxolutein against Glutamate-Induced Neuronal Damage"

_ijms, 2023, doi:10.3390/ijms241512008_

Round 1
Reviewer 1 Report
This submitted manuscript investigated the protective effects of 3’-epilutein and 3’-oxolutein against gluta- 2 mate-induced neuronal damage. By using SH-SY5Y neuroblastoma cells, the authors did a lot of biochemical studies. Overall, the manuscript was in good structure and straightforward, and the findings were interesting. The reviewer has some concerns on this manuscript.
1) The abstract mentioned about the viability. But there was no such data on the result part.
2) Figure legends were too lengthy. Some of the description should be put into main context.
3) The figure 4 and 6 had no control columns.
4) The whole figures were not made in a standard format. The authors may use Prism software for better presentation.
5) There were only mRNA results in Figure 6. However, the authors should present protein data (WBs) to validate their findings.
Should be improved.
Author Response
Answers for Reviewer 1.
This submitted manuscript investigated the protective effects of 3’-epilutein and 3’-oxolutein against glutamate-induced neuronal damage. By using SH-SY5Y neuroblastoma cells, the authors did a lot of biochemical studies. Overall, the manuscript was in good structure and straightforward, and the findings were interesting. The reviewer has some concerns on this manuscript.
1) The abstract mentioned about the viability. But there was no such data on the result part.
We apologize for the mistake. The mention of viability measurement has been removed from the abstract. The results of the viability measurements are presented in the Supplementary Figures (Figure S1-S3).
2) Figure legends were too lengthy. Some of the description should be put into main context.
Thank you for your comment. The figure legends have been shortened. The removed descriptions can be found in the Materials and Methods section.
3) The figure 4 and 6 had no control columns.
Thank you for your comment. The relative expression of the controls was regarded as 1 and is not indicated on the column diagrams to avoid cluttered figures. The description “The relative expression of the controls was regarded as 1 and is not indicated on the column diagrams.” was added to the figure legends of Figures 4 and 6 and it can be found in the Materials and Methods section (4.9.).
4) The whole figures were not made in a standard format. The authors may use Prism software for better presentation.
Thank you for your advice. We do not have full access to Prism software. The figures have been improved and standardized. We decided to sign the significance with an asterisk and cross avoiding the cluttered figures, both with a p < 0.05 level of significance.
5) There were only mRNA results in Figure 6. However, the authors should present protein data (WBs) to validate their findings.
Thank you for your comment. Due to the lack of specific antibodies we choose to determine the thiol concentration.
Antioxidant compounds provide protection for cellular lipids, proteins, and nucleic acids from peroxidative damage. Thiols hold significant importance in biological processes because of their ability to interact with free radicals and their potent reducing capabilities. [PMID: 12398152; 10.5772/intechopen.96682] The level of thiol compounds can play a major role in modulating lipid peroxidation intracellularly. Thiol compounds, such as glutathione and cysteine, are important antioxidants present within cells. These compounds contain thiol (-SH) groups, which are highly reactive and can undergo reversible oxidation-reduction reactions. [10.1016/j.freeradbiomed.2019.05.035] Thiol compounds act as antioxidants by donating electrons to free radicals and ROS in the cell. Through the neutralization of ROS, thiol compounds help protect cellular structures from oxidative damage and lipid peroxidation. [10.5772/intechopen.96682] Thiol compounds, in particular glutathione, play a critical role in the inhibition of lipid peroxidation by removing lipid peroxides and the reactive intermediates formed during the process thus providing protection for the cells. [10.1016/s0753-3322(03)00043-x] Thiol antioxidants have been shown to protect against glutamate-induced cytotoxicity. [10.1152/ajpregu.1997.273.5.R1771; PMID: 2778712] On the other hand, excessive ROS production or oxidative stress increases the demand for thiol compounds such as GSH. Under these conditions, cellular and serum concentrations of thiol compounds may be decreased or depleted, leading to a reduced antioxidant capacity. Therefore, lipid peroxidation may become more frequent and contribute to cell damage. [10.1016/j.freeradbiomed.2019.05.035; 10.1016/j.cotox.2018.03.005; 10.1097/00002030-199714000-00005] In our results, a decrease was detected in the concentration of thiols upon glutamate treatment while 3’-epilutein and 3’-oxolutein amended the reductions. We suppose that the gradual decrease of thiol concentration that was observed upon glutamate is due to the activation of lipid peroxidation. The utilization of 3’-epilutein and 3’-oxolutein restored or increased thiol levels proposing they prevent and/or inhibit lipid peroxidation mediated by glutamate.
Thiol compounds also act as important co-factors for certain enzymes, helping to regulate biochemical processes within cells. Additionally, some thiols serve as signaling molecules in cellular communication and play roles in regulating immune responses and other physiological functions. [10.1016/j.bbagen.2012.11.020; 10.1016/s0070-2137(01)80001-7] Nrf2 is an emerging regulator of cellular resistance to oxidants including thiol–based redox signaling. In oxidative stress, the thiol groups in Keap-1 are oxidized inducing the dissociation of Nrf2, which then translocates into the nucleus and binds to the promoter region of antioxidant-responsive elements followed by the induction of the transcription of different antioxidant genes. [10.1146/annurev-pharmtox-011112-140320; 10.1089/ars.2016.6925] Thioredoxins and glutathione play an essential role in the maintenance of redox balance in CNS. In this regulation, the organisms are equipped with various antioxidant enzymes, including superoxide dismutase, catalase, glutathione peroxidase, and peroxiredoxin, and small antioxidants, such as glutathione, coenzyme Q, vitamin C and E. [10.1089/ars.2016.6925] In our results, glutamate decreased the level of small molecule antioxidants which was compensated by 3’-epilutein treatments besides an increase was detected in the presence of 3’-epilutein alone as well. Glutamate reduced SOD activity in the cells while 3’-epilutein increased SOD activity in each treatment and 3’-oxolutein was also able to modify SOD activity in the OG5 treatments.
In addition, Nfr2 is able to modify iron homeostasis in cells. As a regulator, Nrf2 induces the transcription of ferritin heavy chain (FTH) to store oxidized iron. [10.3389/fvets.2018.00242] Accumulation of iron is a participant in ferroptosis and lipid peroxidation as well as oxidative stress. Ferroptosis is characterized by iron-dependent ROS accumulation, glutathione depletion, and ultimately lipid peroxidation. [10.1016/j.freeradbiomed.2018.09.043; 10.4103/1673-5374.358614; 10.1038/s41419-021-04490-1; 10.1038/s41419-020-2298-2] We measured an increase in the mRNA expression of TfR1 and FTH and an accumulation of total iron levels in SH-SY5Y cells upon glutamate treatments.
There is a delicate balance between thiol compounds and lipid peroxidation within cells. Thiol compounds serve as pivotal antioxidants that protect cells from lipid peroxidation and related damage. However, in conditions of oxidative stress or thiol depletion, lipid peroxidation can be exacerbated, leading to cellular dysfunction, and potentially contributing to various diseases and aging processes. [https://doi.org/10.1016/bs.vh.2022.09.002] ALOX5 and ALOX15 are major contributions to generating lipid peroxides, while ACSL4 plays a role in the development of lipid peroxidation and ferroptosis [59]. [10.1021/acscentsci.7b00589; 10.1155/2019/2749173; 10.1016/j.plipres.2018.11.001; 10.1038/s41556-021-00818-3] The activation of ACSL4 can lead to lipid ROS-driving ferroptosis, in addition lipids can be degraded by ALOXs which is related to an increase in iron levels. [10.3389/fcell.2020.590226; 10.1631/jzus.B2200194; 10.3389/fcell.2020.586578]
Although we did not perform protein measurements of ALOX5, ALOX15, and ACSL4, the alteration of the examined gene expression is presumed to reflect the initiation of transcriptional and translational changes which results can be seen in the activation of ROS, changes in thiol compounds, in total antioxidant capacity and in the level of iron in the cells.
Comments on the Quality of English Language
Should be improved.
Thank you for your comment. We improved the quality of the manuscript.

Reviewer 2 Report
This is a well written paper, and the experimental design is very good. I thought the authors should have introduce more info on glutamate earlier in the paper. The author stated that leutein is likely neuroprotective, but go on to explain that the oxidation of lutein metabolization is responsible for the neuroprotection, probable should have been explained more . Despite this tha paper well done and referenced.
Author Response
Answers for Reviewer 2.
This is a well written paper, and the experimental design is very good. I thought the authors should have introduce more info on glutamate earlier in the paper. The author stated that leutein is likely neuroprotective, but go on to explain that the oxidation of lutein metabolization is responsible for the neuroprotection, probable should have been explained more . Despite this tha paper well done and referenced.
Thank you for your kind review. The Introduction section has been completed with information on glutamate and the action of lutein.

Round 2
Reviewer 1 Report
The authors have addressed all concerns raised by the review.